# Effect of Different Crowding Agents on the Architectural Properties of the Bacterial Nucleoid-Associated Protein HU

**DOI:** 10.3390/ijms21249553

**Published:** 2020-12-15

**Authors:** Szu-Ning Lin, Gijs J.L. Wuite, Remus T. Dame

**Affiliations:** 1Leiden Institute of Chemistry, Leiden University, 2333 CC Leiden, The Netherlands; s.lin@vu.nl; 2Department of Physics and Astronomy, Vrije Universiteit Amsterdam, 1081 HV Amsterdam, The Netherlands; 3LaserLaB Amsterdam, Vrije Universiteit Amsterdam, 1081 HV Amsterdam, The Netherlands; 4Centre for Microbial Cell Biology, Leiden University, 2333 CC Leiden, The Netherlands

**Keywords:** HU, nucleoid-associated protein, bacterial chromatin, nucleoid, crowding, protein-DNA interaction, TPM

## Abstract

HU is a nucleoid-associated protein expressed in most eubacteria at a high amount of copies (tens of thousands). The protein is believed to bind across the genome to organize and compact the DNA. Most of the studies on HU have been carried out in a simple in vitro system, and to what extent these observations can be extrapolated to a living cell is unclear. In this study, we investigate the DNA binding properties of HU under conditions approximating physiological ones. We report that these properties are influenced by both macromolecular crowding and salt conditions. We use three different crowding agents (blotting grade blocker (BGB), bovine serum albumin (BSA), and polyethylene glycol 8000 (PEG8000)) as well as two different MgCl_2_ conditions to mimic the intracellular environment. Using tethered particle motion (TPM), we show that the transition between two binding regimes, compaction and extension of the HU protein, is strongly affected by crowding agents. Our observations suggest that magnesium ions enhance the compaction of HU–DNA and suppress filamentation, while BGB and BSA increase the local concentration of the HU protein by more than 4-fold. Moreover, BGB and BSA seem to suppress filament formation. On the other hand, PEG8000 is not a good crowding agent for concentrations above 9% (*w*/*v*), because it might interact with DNA, the protein, and/or surfaces. Together, these results reveal a complex interplay between the HU protein and the various crowding agents that should be taken into consideration when using crowding agents to mimic an in vivo system.

## 1. Introduction

In 1975, HU (Histone-like protein from strain U93) was discovered as one of the most abundant proteins in *Escherichia coli* (*E. coli*) cells [1]. *E. coli* HU is a small (9 kDa) DNA binding protein encoded by two different genes, *hupA* and *hupB*. The products of these genes assemble into hetero- and homo-dimers in solution (HU_αβ_, HU_αα_ and HU_ββ_) (18 kDa) [2]. While heterodimers exist in *E. coli*, homodimers are most common in other bacteria. The protein is a nucleoid-associated protein (NAP), which is a family of proteins involved in the organization and compaction of genomic DNA as well as the regulation of genomic DNA transactions [3,4,5]. HU exhibits two DNA binding modes: a DNA bending mode [3,4] and an extension mode with HU filaments forming on DNA [6,7,8,9,10] (Figure 1A,B). The structures of HU alone and HU–DNA complexes have been determined using X-ray crystallography and NMR [3,4,11]. The protein consists of an “arm” composed of β-sheets and a “body” composed of α-helices. In the bending mode, the two “arms” of the dimer insert into the minor groove of DNA [6] inducing two bends spaced by 9 bps. It is unclear how HU dimers interact in filaments. However, it has been suggested that in filaments, one of two “arms” of a dimer interacts across DNA with another “arm” of another dimer, and the other “arm” of each dimer interacts with the “body” of the neighboring dimer [9] (Figure 1B). The transition between the two binding modes is dependent on HU concentration [7,12] and potentially also on physicochemical conditions.

Macromolecules, usually polymeric molecules, with a molecular mass greater than a few thousand Dalton, cause diverse effects in a space confined system. Macromolecular crowding effects arise because of macromolecules’ occupancy of a significant fraction in the confined volume, which consequently drives changes in the conformation of proteins [14], diffusion [15] of proteins, and protein–protein interaction [16], concomitantly enhancing and perturbing protein binding energy and affinity to DNA [17]. A demonstration of this effect is the seminal study by Murphy and Zimmerman [18], which reports on the effects of macromolecular crowding on HU binding to DNA, using polyethylene glycol (PEG8000) as a crowding agent. In their study, PEG8000 was shown to enhance the DNA binding affinity of HU. Another nucleoid-associated protein, H-NS, has also been investigated in the presence of PEG. That study reveals that that the binding cooperativity of H-NS proteins was increased [19].

Ionic strength also affects the binding of HU to DNA. The effects of ionic strength on proteins have been investigated for many years [20]. Ionic strength has been shown to affect DNA binding affinity as a consequence of charge shielding on the interaction partners [21]. In line with generic observations, high NaCl concentration has been found to reduce HU binding affinity [22,23]. Salt conditions affect the transition between the two binding modes of HU because filamentation requires a high HU density on DNA, which is harder to achieve at high salt conditions.

Murphy and Zimmerman showed that the condensation of DNA by HU is significantly increased by increasing concentrations of PEG8000 and MgCl_2_ [18]. Inspired by this earlier study as well as by indications from other studies showing that different crowding agents have different effects on protein binding [24], we systematically investigated the effect of different crowding agents on the binding of HU to DNA. We selected two types of crowding agents from milk, blotting grade blocker (BGB) and bovine serum albumin (BSA), as well as polyethylene glycol 8000 (PEG8000)—a synthetic polymer—to investigate the impact of crowding. BGB is a non-fat milk-product mixture of predominantly casein micelles containing various large globular proteins ranging from 50 to 600 nm in diameter [25,26]. It is commonly used to block non-specific membrane binding in Western blotting and as a surface passivation agent in single molecule assays [10]. It consists of neutrally charged proteins that likely do not exhibit charge interactions with DNA and HU. BSA (66.3 kDa [14]) is a protein that is commonly used as a crowding agent in biochemical reactions. However, it exposes negative charges on its surface, and it might interact with DNA and proteins [27]. PEG has been used in many studies on crowding [17,19]. PEG8000 (8 kDa), used in this study, is not an ideal crowding agent, as it is known to interact with proteins at higher concentrations [28]. In dilute conditions, PEG8000 attains a globular shape with a neutrally charged surface [29]. However, it is unknown whether, and if so at which concentration, PEG8000 interactions with HU occur, and how this affects HU. We also investigate the effect of Mg^2+^ in our experiments on crowding. Understanding the effects of magnesium ions is important because of their essential role in the functionality of many proteins and enzymes [29,30,31,32,33,34]. For our studies on HU binding to DNA, we make use of tethered particle motion (TPM) assays, in which the x–y movement of tethered microspheres is recorded (Figure 1C–top). The root-mean-square (RMS) displacement obtained from an x–y position distribution is used for characterizing the physical properties of protein–DNA complexes. A decrease in RMS indicates a compacted protein–DNA structure as in Figure 1C (red distribution), and an increase in RMS indicates an extended protein–DNA structure as in Figure 1C (blue distribution).

## 2. Materials and Methods

### 2.1. Protein Expression and Purification

The purification was carried out as described before [35]. HU was overproduced in *E. coli* KA 1790. Cells were grown on LB in 0.5 L batch culture at 30 °C. At an $OD_{600}$ of 0.6, over production was introduced by the addition of 0.5 l LB (pre-heated to 60 °C), and incubation was continued at 42 °C. Two hours after induction, cells were collected and the pellets were frozen. Then, pellets were resuspended in HB buffer (25 mM Tris, pH 8.0, 1 mM EDTA, 5 mM β-mercaptoethanol, 10% glycerol) with 250 mM NaCl and further lysed by sonication. The lysate was cleared by centrifugation at 35,000 r.p.m. for 30 min at 4 °C. Subsequently, the supernatant was incubated for 20 min at 90 °C. The lysate was cleared from heat-denatured proteins by centrifugation at 10,000 r.p.m. for 15 min. Then, the supernatant was diluted with HB buffer to a NaCl concentration of 100 mM and loaded onto a P11-column pre-equilibrated with the same buffer. A linear NaCl gradient up to 1 M was applied, and HU was eluted at 400–500 mM NaCl. HU-containing fractions were pooled, diluted with HB buffer to a NaCl concentration of 50 mM NaCl, and loaded onto a Resourse S-column pre-equilibrated with the same buffer. A linear NaCl gradient up to 750 mM was applied and HU was eluted around 200 mM NaCl. The purity of the HU protein was verified on an SDS-PAGE gel and quantified using BCA assay (Thermo scientific). DNA binding activity was confirmed by TPM.

### 2.2. DNA Substrate

An end-labeled DNA substrate of 685-bp (53% GC content) [12] was obtained by PCR using biotin–and digoxygenin (DIG)-labeled primers and plasmid pRD118 (laboratory collection) as a template. pRD118 was constructed by inserting a 685 bp long fragment from the *S. solfataricus* P2 genome into the NdeI and BamHI site of pET3-his. The plasmid and plasmid sequence are available upon request. The PCR product was purified using the GenElute PCR Clean-up kit (Sigma-Aldrich, Zwijndrecht, Netherlands)).

### 2.3. Beads

Streptavidin-coated polystyrene beads (0.44 µm in diameter; Kisker Biotech, Steinfurt, Germany) were diluted in 0.01% (*w*/*v*) in buffer A (10 mM Tris-HCl (pH7.5), 150 mM NaCl, 1 mM EDTA, 1 mM DTT, and 3% (*w*/*v*) glycerol) and bath sonicated [36]. Additional DIG-coated polystyrene beads (0.9 µm in diameter; Spherotech) [37] were added after sonication as reference beads settled on the surface. The DIG-coated beads will bind to the surface and serve as reference beads for drift correction.

### 2.4. Flow Cell

Two sizes of cover glasses (30 mm and 28 mm in diameter; Thermo Fisher Scientific, Bleiswijk, Netherlands) were first sonicated in acetone and then sonicated in ethanol for cleaning. The parafilm strips cut by scissors were aligned using a tweezer and sandwiched between the two different sizes of a cover glass. The parafilm in between the glasses was melted on a heating plate at 80 °C and formed the channels at room temperature. A homogeneous force (200 g in weight) was applied to the flow cell during cooling. In total, 8–9 chambers were made on each pair of cover glasses, each with a volume of ≈15 µL.

### 2.5. Tethered Particle Motion Experiments

For TPM, a streptavidin-coated microsphere was attached to a double-stranded DNA (dsDNA), which was labeled separately with biotin and digoxigenin at two ends. Biotin binds to the streptavidin-coated beads, and digoxigenin binds to the anti-digoxigenin antibodies with which the inner surface of the flow cell was coated. The TPM instrument used and the experimental approach were described elsewhere [36]. In short, surface-tethered beads were illuminated with an LED, imaged using a Nikon oil immersion lens (100×) and imaged with a CCD camera. Using a bead tracking program written in LabView [38], we determined the two-dimensional motion of the tethered beads and we can calculate their root-mean-square (RMS) displacement. The RMS is a proxy of the conformation of the HU–DNA complex.

Flow cells were incubated with 20 µl/mL anti-DIG antibodies (Roche) for 10 min. Passivation of the surface was achieved by incubating buffer B (0.2% (*w*/*v*) BSA (Sigma-Aldrich, Zwijndrecht, Netherlands) in phosphate-buffered saline (PBS) (137 mM NaCl, 2.7 mM KCl, 10 mM Na_2_HPO_4_, 1.8 mM KH_2_PO_4_, pH 7.5)), then buffer C (0.5% (*w*/*v*) Pluronic (Sigma-Aldrich) in PBS), and then buffer D (buffer A with the addition of 4% (*w*/*v*) BGB (Bio-Rad)). Each buffer incubation lasted 10 min. An amount of 100 pM DNA was introduced into flow cells after flushing buffer A and was incubated for 10 min.

Microspheres (0.44 and 0.9 µm in diameter) suspended in buffer A were introduced into flow cells after DNA incubation, and the free DNA were flushed out with buffer D. After 10 min incubation, free beads were removed by flushing measuring buffer in a volume of 80 µl. Then, the flow cells were ready to be used. Samples are a titration series from 0 to 1600 nM HU using desired measuring buffers: 0.5% and 1% (*w*/*v*) BGB, 1.25%, 5%, 10% (*w*/*v*) BSA, 3%, 9%, 15% (*w*/*v*) PEG8000 in buffer I, the absence of MgCl_2_, (10 mM Tris (pH 7.5), 50 mM NaCl, 1 mM EDTA), and for the experimental condition containing MgCl_2_: 0.5% (*w*/*v*) BGB, 10% (*w*/*v*) BSA, and 3% (*w*/*v*) PEG in buffer II, the presence of MgCl_2_, (10 mM Tris (pH 7.5), 50 mM NaCl, 8 mM MgCl_2_).

### 2.6. Data Analysis

The RMS value of the excursion of each individual bead was calculated from *x*- and *y*-coordinates of a 40 s time trace (drift corrected by the 0.9 µm stuck beads) as:(1)$$RMS=\sqrt{{\left(x-\bar{x}\right)}^{2}+{\left(y-\bar{y}\right)}^{2}}$$
where $\bar{x}$ and $\bar{y}$ are averaged over the full-time trace. The symmetry of the excursion of the tethered beads was evaluated by calculating the anisotropic ratio a=lmajorlminor from the *xy*-scatter plots, where $l_{major}$ and $l_{minor}$ represent the major and minor axis of the *xy*-scatter plot, respectively. Only tethers with symmetry a≤1.3 were selected for further analysis. For each measurement condition, RMS values corresponding to each HU concentration were obtained from the fitting of a single Gaussian function to the pooled RMS values of individual tethers (N > 70, see Appendix A).

### 2.7. Quantitation of DNA Coverage and Calculation of DNA Binding Properties

To compare quantitatively the binding affinities of proteins at the different conditions, we calculated the fractional coverage of HU on DNA, υ, as follows [39]:(2)$$\upsilon =\frac{\sqrt{L_{p}{}^{-1}}-\sqrt{L_{p,bare}{}^{-1}}}{\sqrt{L_{p,saturated}{}^{-1}}-\sqrt{L_{p,bare}{}^{-1}}}$$
where *L_p_* represents the measured persistence length, *L_p,saturated_* represents the minimum persistence length at saturation, and *L_p,bare_* represents the persistence length of bare DNA. The persistence length is converted from *RMS* using the relationship determined elsewhere [12] as:(3)RMS=233−156(1+0.086·Lp)0.45

In this approach, it is assumed that each bound protein gives an equal contribution to the decrease in DNA stiffness. To calculate binding affinities under the different conditions, the Hill function was fitted to the fractional coverage [40,41]:(4)$$f\left(x\right)=\upsilon_{0}+\left(\upsilon_{max}-\upsilon_{0}\right)\frac{x^{n}}{x^{n}+k^{n}}$$
where *k* represents the Michaelis constant, and *n* represents the cooperativity factor.

## 3. Results

### 3.1. MgCl_2_ Enhances Compaction of DNA by HU

To investigate the effect of divalent cations on HU binding, we incubated DNA with HU in a buffer either without or with 8 mM MgCl_2_. The protein was titrated over a concentration range in which we expect to reproduce both binding regimes, as seen in earlier work [7,12]. In the absence of magnesium ions, the root-mean-square displacement (RMS) of the movement for bare DNA was found to be 140.8 ± 0.2 nm (N = 300, standard error of the mean (SEM)); Figure 2A gray horizontal line). At an HU concentration of 150 nM, at which the most compacted state was reached, the RMS was found to be 102.4 ± 0.5 nm (N = 198). In the presence of MgCl_2_, the RMS of bare DNA was found to be 140.3 ± 0.3 nm (N = 302), which is the same as that observed without MgCl_2_. However, the RMS in the most compact state in the presence of MgCl_2_ was found to be 97 ± 1 nm (N = 222), which is 5% lower than that in the absence of MgCl_2_ (Figure 2A). Thus, our observations suggest that MgCl_2_ enhances the compaction of DNA by HU. The most compact state was observed at 150 nM HU both in the absence and presence of MgCl_2_, which indicates that MgCl_2_ does not alter the DNA binding affinity of HU.

Interestingly, we found two RMS populations for data obtained at HU concentrations between 150 and 400 nM (Figure 2A and Appendix A). One population corresponds to the reported compact state, while the other population has a high RMS close to that of bare DNA. This phenomenon was independent of the presence of MgCl_2_. As HU dimers are known to bind cooperatively, it could be that in a subset of complexes, HU–DNA fragments not covering the entire length of the DNA are formed, thus yielding a larger RMS. Structurally, this would correspond to the co-existence of the two binding modes; we will refer to it as the “co-existence state” (Figure 2B).

### 3.2. BGB Increases Local HU Concentration

We investigated the effect of BGB on HU binding at various conditions: 0.5% and 1% (*w*/*v*) of BGB in the absence of MgCl_2_, and 0.5% BGB in the presence of MgCl_2_. In the presence of BGB, gradual compaction and de-compaction over the tested concentration range are observed (Figure 3A), which is in contrast with the much more abrupt switching between binding regimes in buffer without BGB. The compaction of HU–DNA becomes notable at 12.5 nM HU concentration in both BGB crowding environments, and the RMS of the tethered beads reduced gradually with increasing HU concentration. In contrast, the decrease in RMS in crowder-free conditions without MgCl_2_ occurs abruptly at a concentration of 150 nM HU. This suggests that BGB increases the local concentration of HU dimer, stimulating HU binding to and the subsequent bending of DNA at much lower HU concentrations. Moreover, in both BGB conditions, the RMS of the most compact HU–DNA state goes down to 99 ± 1 nm (N = 173; 0.5% (*w*/*v*) of BGB) and 97.0 ± 0.5 nm (N = 191; 1% (*w*/*v*) BGB), which is ≈4% lower than in the crowder-free case (102.4 ± 0.5 nm). This reduction suggests that BGB also slightly enhances the extent of HU–DNA compaction.

There are some notable differences between the impact of the two BGB concentrations as well. First, the maximal compaction is reached at a lower HU concentration (25 nM versus 100 nM) in low BGB concentration compared to the higher BGB concentration. This observation suggests that there is a balance between the crowding effect on binding energy and on diffusion i.e., it might be that too much crowding by BGB hinders HU access to DNA. Second, at higher HU concentrations where HU forms filaments along the DNA, the RMS increases less in the presence of BGB compared to conditions without BGB. Moreover, the impact on filament formation in 1% BGB is stronger than in 0.5% BGB; see the difference in slope in Figure 3A in the filamentation regimes. BGB might inhibit HU filamentation. This could be due to changes in the size of BGB, which increases the repulsive force between the protein and crowding agent [42], and it consequently reduces the probability that multiple HU dimers bind to DNA at the same time. The repression of filamentation by BGB might also explain why we do not observe the two populations as found in the condition without crowding agent (Figure 2A).

When we compare the effects of this crowding agent in the presence of MgCl_2_, we observe that the most compact state appears at 200 nM with an RMS of 90.3 ± 0.4 (N = 220) nm, which is a reduction by 7% compared to the RMS without BGB (97 ± 1 nm, N = 222) (Figure 3B). Moreover, given that the compaction of DNA by HU in the presence of MgCl_2_ is already enhanced (see Figure 2A), it can be concluded that MgCl_2_ works synergistically with BGB enhancing DNA compaction. At low HU concentration, the trend in reduction of RMS in 0.5% BGB with MgCl_2_ is similar to that in 0.5% BGB without MgCl_2_. At high HU concentration, double the HU concentration is needed in the presence of MgCl_2_ to reach the same level of filament formation, indicating that BGB with MgCl_2_ suppresses filament formation.

### 3.3. BSA Increases Local HU Concentration

To investigate the impact of BSA on HU binding, we used three concentrations (1.25%, 5% and 10% (*w*/*v*)) of BSA in the absence of MgCl_2_, and 10% BSA in the presence of MgCl_2_. In the presence of BSA, compaction and de-compaction occur across a larger concentration range compared to the condition without BSA (Figure 4A). The compaction becomes visible at 6.25 nM HU in 1.25% and 5% BSA, and at 12.5 nM HU in 10% BSA, whereas the reduction in RMS in crowder-free condition occurs abruptly at 150 nM HU. This suggests that BSA increases the local concentration of HU dimer. In 5% and 10% BSA conditions, the most compact states are 101.8 ± 0.5 nm (N = 441) and 101 ± 1 nm (N = 224) at 50 nM HU while at 1.25% BSA, it is 105 ± 1 nm (N = 183) at 25 nM HU. Based on these observations, we can conclude the following: (1) the most compact state attained in the presence of BSA does not show a significant difference compared to that in absence of BSA. (2) A too high BSA concentration (above 5%) is less effective in increasing the local HU concentration. As mentioned above in Section 3.2, the crowding agent may interfere with HU binding either by steric hindrance or by interacting with HU and/or the DNA [27], thus indirectly affecting the binding affinity of HU to DNA. Finally, as is the case for BGB, BSA also suppresses the co-existence of two states in the intermediate concentration range (Figure 4A). Moreover, increasing the concentration of BSA from 1.25% to 5% suppresses the filamentation, as seen for BGB. In general, the regulation of filament formation occurs in different ways: it can be modulated by the buffer environment but also by the DNA substrate [43]. Since BSA can potentially interact with DNA [44], the properties of DNA might in our case be affected, reducing the ability of HU to form filaments.

Next, we compare the effects of this crowding agent in the presence of MgCl_2_. In the presence of MgCl_2_ and 10% BSA, the most compact state has an RMS of 86.8 ± 0.3 (N = 284) nm at 800 nM, which is a reduction of ≈11% compared to the RMS in presence of MgCl_2_ without BSA (97.3 ± 0.8 nm, N = 222) (Figure 4B). In addition to the enhanced compaction, the most compacted state occurs at a higher HU concentration. This phenomenon cannot be explained as due to the impact of MgCl_2_ or BSA alone; BSA together with MgCl_2_ yields more stable compaction, which is achieved at higher HU concentration. The effect of salt concentration, NaCl, on BSA has been investigated by Yoshikawa et al. [27], who showed that the size of BSA increases in low salt. Thus, in our experiments with MgCl_2_, BSA could be reduced in size, resulting in a lower effective volume exclusion effect. However, the BSA change in terms of size and surface charge can still possibly inhibit filamentation by interacting with the DNA and thus preventing multiple HU dimers to bind in a filament on DNA.

### 3.4. PEG8000 Enhances HU Compaction

In order to investigate the impact of PEG8000 on HU binding, we used three concentrations (3%, 9%, and 15% (*w*/*v*)) of PEG8000 in the absence of MgCl_2_ and 3% PEG8000 in the presence of MgCl_2_. The presence of PEG8000 causes gradual compaction and de-compaction across a larger concentration range compared to the condition without PEG8000 (Figure 5A). The compaction starts at 12.5 nM HU in 15% PEG, 50 nM HU in 9% PEG, and 100 nM HU in 3% PEG, whereas the reduction in RMS in crowding-free condition occurs at 150 nM HU. This suggests that with a high enough concentration, PEG8000 is able to increase the local HU concentration. We observed no significant effect on the value of the lowest RMS along the titration curve in 3% and 9% PEG, but 15% PEG yields a dramatic 30% decrease in RMS (71 ± 1 nm, N = 210) compared to the crowder-free condition. It has been observed that PEG8000 polymers interact with themselves in a high concentration [45]. These polymers can form a larger ball-shape structure: the hydrophobic chains of polymers face inward, and the hydrophilic OH ends of the chains face outward. It is possible that the DNA gets covered by PEG8000, leading to an extended configuration; the RMS of bare DNA in 15% PEG8000 is higher compared to the condition without PEG8000 (Figure 5A, pink horizontal line). This extended DNA may favor HU binding, meaning that more HU dimers can bind and cause stronger compaction.

Next, we investigated the effect of this crowding agent in the presence of MgCl_2_. HU dimer was diluted in the buffer containing 8 mM MgCl_2_ and 3% PEG8000. In these conditions, the most compact state appears at an RMS of 90 ± 2 (N = 229) nm at 100 nM, which is a reduction by 7% compared to the RMS in the presence of MgCl_2_ without PEG8000 (97 ± 1 nm, N = 222) (Figure 5B). In contrast, the most compact state in the absence of MgCl_2_ does not change regardless of the PEG8000 concentration. These results indicate that MgCl_2_ works synergistically with PEG8000 in enhancing DNA compaction. Moreover, the PEG and MgCl_2_ combination seems to increase the local HU concentration with the most compact state appearing at 100 nM HU.

Note that the RMS of bare DNA in 9% and 15% PEG show significantly higher values (Figure 5A, green and pink horizontal lines) compared to the bare DNA in the absence of MgCl_2_ and crowder-free condition (Figure 5A, gray horizontal line). This suggests that PEG8000 at high concentrations interacts with DNA (Appendix A). It was noted by Zhou et al. [46] that PEG8000 cannot be described quantitatively in terms of excluded volume alone; its effects on different proteins differ strongly. Moreover, in the presence of PEG, we observe often two tether populations (Figure 5B). The appearance of these populations is similar to what we observe in crowding-free conditions (Appendix A). Finally, at high PEG8000 and high HU concentration, DNA tethered beads get stuck on the channel surface. Taken all the observations at a high concentration of PEG8000 together, we conclude that PEG8000 might not simply exclude volume but also interact with the DNA, protein, and/or surfaces.

### 3.5. HU Binding Cooperativity in Crowded Conditions

To compare the effect of all the different crowding agents and the impact of MgCl_2_, we plotted the RMS values of the most compact state as a function of HU concentration in Figure 6. BSA (solid circles) yields the strongest effect in volume exclusion; even low amounts of BSA strongly increase the local HU concentration on the DNA, which directly results in DNA compaction. For all the experimental crowding conditions, the most compact state is shifted to higher HU concentration in the presence of MgCl_2_. The trend that the RMS becomes lower when a higher HU concentration is needed to achieve maximum compaction (Figure 6) suggests that the combination of MgCl_2_ and crowding agents suppresses filament formation/propagation, increasing the opportunity for HU to bend and compact the DNA more. The combination of MgCl_2_ and BSA yields the largest degree of compaction for all conditions tested. It is known that MgCl_2_ influences the surface charge and the size of BSA and thus, in the presence of MgCl_2_, it seems the strongest in preventing multiple HU dimers to form filaments, yielding the most compacted HU–DNA complexes.

To get more insight into how HU dimers induce stronger compaction in the presence of MgCl_2_ and crowders, we estimate the amount of bound protein, which we calculate from the fraction of DNA covered by HU [39] using the apparent persistence length (Lp), which is derived for each RMS value using a numerical approximation by simulation [12] (see Materials and Methods). We found that HU coverage, the fraction of DNA that is covered by protein, in the absence of crowding agent shows a sharp transition in the compaction mode. The fraction of DNA covered by HU increased from 10% to 100% when increasing HU concentration from 100 to 150 nM (Appendix A). DNA coverage in crowding conditions more gradually increases with increasing HU concentration.

Then, we obtain the cooperativity of HU binding [39] from these coverage data. We extracted this binding parameter by fitting a Hill function [40] to the curves of fractional coverage of DNA as a function of HU concentration. Using the Hill function here is not to pursue an accurate number but rather to facilitate comparison. We divided the binding curves into a compaction regime and an extension regime using the most compact state as the boundary between the two regimes. Next, we analyzed both regimes independently. In both the presence and absence of MgCl_2_ in crowder-free conditions, the compaction regime exhibits an extremely sharp transition (Figure 2A). It is not possible to obtain a Hill curve fit from the limited number of data points. Therefore, we implement three sets of fitting curves (Appendix A) as a reference and estimate the cooperativity of HU in the presence and absence of MgCl_2_ in the compaction regime. The estimated cooperativity is above 20. For 10% BSA in the presence of MgCl_2_, we obtained the estimated value using the same method due to the limited number of data points (Figure 7, bottom empty diamond). In the presence of nearly all crowding agents, this sharp transition is suppressed, i.e., crowding agents make compaction a less cooperative process. Compared to the low cooperativity in the compaction regime (Figure 7, top), enhanced cooperativity in the extension regime is found in the presence of MgCl_2_ (Figure 7, bottom).

PEG8000 affects the DNA compaction and extension dynamics in a manner distinctly different from BSA and BGB. In fact, PEG8000 likely acts directly on the DNA as well as on the tethered beads and chamber surface. This might affect the DNA configuration measurement and dramatically lower the RMS values. Based on its anomalous behavior, we propose that PEG8000 is not a suitable crowding agent for studying HU–DNA interactions.

## 4. Discussion and Conclusions

It has been appreciated for a long time that macromolecular crowding affects cellular processes, but it is difficult to reproduce these crowding conditions in an in vitro setting. One approach is to use cell extracts [47], but while these may be more “physiological” in terms of composition, composition is often less well defined. Thus, reconstitution with purified components and added, defined crowding agents is an important complementary approach. So far, there is still no universal crowding agent to appropriately mimic intracellular crowding conditions for different types of protein systems. As a result of the size, shape, concentration [46,48], and surface charge of crowding agents, crowding agents are not always generating “pure” crowding effects on the proteins of interest. Based on our TPM studies, we reveal that three types of crowding agents (BGB, BSA, PEG8000) have a distinct impact on the dual-binding modes of HU (compaction and extension/filament formation). Our results indicate that inside a cell, due to the crowded environment, regarding the binding of individual HU dimers, promoting compaction is biased over filament formation along DNA, which would not contribute to DNA compaction in any obvious manner.

We have observed an enhanced condensation of HU–DNA in the presence of MgCl_2_, but we did not observe an increased affinity/stability of HU binding to DNA as detected using SDS–polyacrylamide electrophoresis [18]. We attribute more value to our observations using TPM in which a stable buffer environment with a constant amount of MgCl_2_ is present, whereas due to the nature of electrophoresis experiments, the environment experienced by protein and DNA is poorly defined in electrophoresis.

We have observed that two states co-exist in the transition phase between condensation and extension in crowder-free conditions. These two distinct populations are absent under crowding conditions. To our knowledge, these two populations are not observed in previous studies. These two populations correspond to either complexes dominated by HU in its bending mode (lower RMS) or complexes dominated by HU forming filaments (higher RMS).

The addition of MgCl_2_ in the presence of crowding agents promotes compaction. This suggests that MgCl_2_ in the presence of crowding agents more specifically suppresses filament formation possibly by changing HU conformation and the structure of crowding agents, leaving more space on DNA for the HU dimer to bend.

To conclude, over the last three decades, in vitro protein studies have mostly been carried out under simple experimental conditions without considering the macromolecular environment. That this environment plays an important role is becoming increasingly clear. Hence, the study of useful crowding agents that mimic such conditions when studying proteins is gaining prominence. Here, we conclude that BGB is a good crowding agent for studying HU–DNA interactions in the presence and absence of MgCl_2_; i.e., it seems the most suitable crowder to increase the HU–DNA interaction. Nevertheless, we showed that the presence of different crowding agents in different buffer conditions can result in vastly different outcomes. Hence, defining suitable crowding conditions to mimic the crowded cellular environment is very hard and probably protein-dependent and thus should be done with utmost care.

An actual intracellular environment is not only crowded but also contains different ions to interact with proteins. Therefore, it will be worthwhile to examine the effect of charged crowding agents interacting with DNA. We have tested how three crowding agents affect one type of protein. This represents only a first step to study the crowding effects in a system. The crowding agent, BGB, is the one that shows a clear volume exclusion effect and is the least influenced by MgCl_2_ in the buffer solution among the crowding agents we have measured. BSA and PEG8000 under certain concentration and ionic conditions can still be an option for mimicking a crowded intracellular environment, but particular care is needed about the effect on the biological functionality of the molecules studies. Resolving the mechanisms underlying all the interactions between crowding agents, proteins, and DNA requires further investigation related to their independent physical and chemical properties.

## Figures and Tables

**Figure 1 ijms-21-09553-f001:**
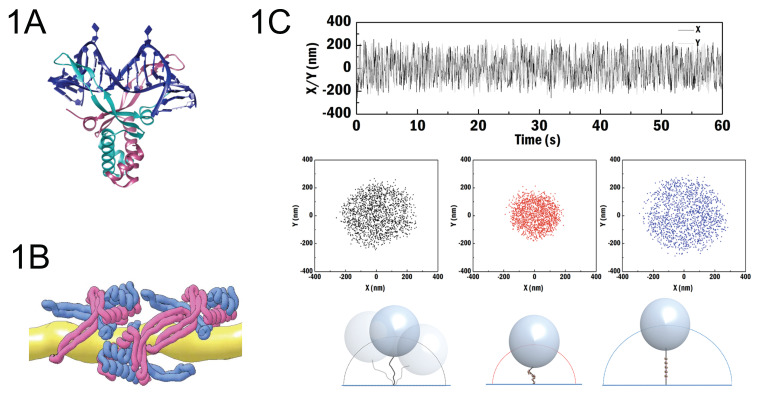
HU exhibits different DNA binding modes with distinct structural effects. (**A**) Model of DNA bending by *Anabaena* HU bound to a pre-distorted dsDNA substrate (X-ray structure of HU–DNA complex, PDB code 1P71 [13]. (**B**) Model of an HU–DNA filament as proposed by Hammel, M. et al. [9] (reproduced under license CC-BY-4.0: https://creativecommons.org/licenses/by/4.0/)). Pink and blue represent HUαβ and DNA in yellow color. (**C**) Typical tethered particle motion (TPM) data obtained for DNA and different types of HU–DNA complexes. Example of a time trace of x and y position of a surface-tethered bead (top). Bare DNA and HU–DNA complexes incubated with 200 nM (bending mode) and 1600 nM (filament formation mode) are shown respectively in black, red, and blue.

**Figure 2 ijms-21-09553-f002:**
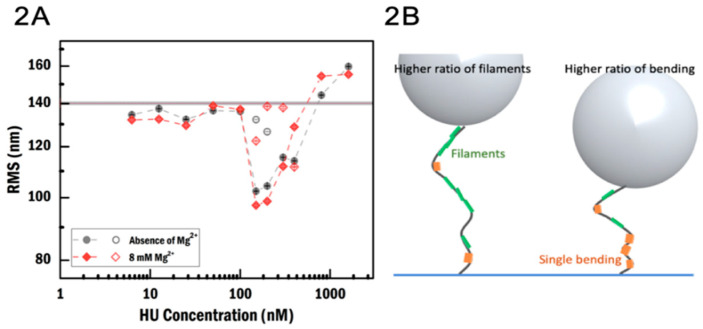
TPM measurements of the effect of MgCl_2_ on binding of HU to DNA. (**A**) Root-mean-square (RMS) as a function of HU concentration in the absence (gray) and presence (red) of MgCl_2_. Open symbols correspond to the smaller subpopulation. The gray horizontal band indicates the RMS of bare DNA in both the absence and presence of MgCl_2_. Error is the standard error of the mean. (**B**) Illustration of the nature of heterogeneous complexes. Depending on the ratio between the binding of individual DNA bending proteins and filament formation, the RMS data exhibit one or two populations.

**Figure 3 ijms-21-09553-f003:**
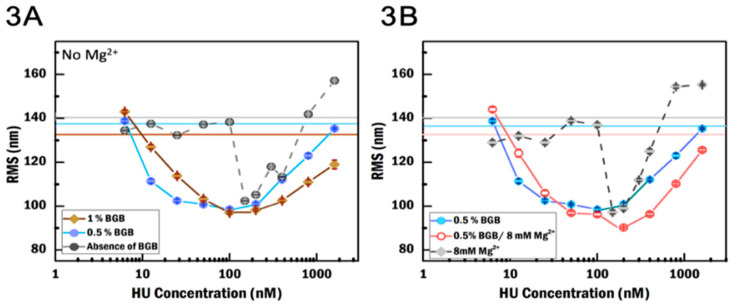
TPM measurements of the effect of blotting grade blocker (BGB) and MgCl_2_ on the binding of HU to DNA. (**A**) RMS as a function of HU concentration in different crowding conditions. Horizontal lines represent the RMS of bare DNA in the absence of BGB (gray), 1% BGB (brown), and 0.5% BGB (light blue) in the absence of MgCl_2_. (**B**) RMS as a function of HU concentration in different crowding conditions and MgCl_2_ concentrations. Horizontal lines represent the RMS of bare DNA in the absence of BGB and in the presence of MgCl_2_ (gray), with 0.5% BGB in the absence of MgCl_2_ (pink) and with 0.5% BGB in the absence of MgCl_2_ (light blue). The second population in RMS both in the absence and presence of MgCl_2_ is removed in (**A**,**B**). Error is the standard error of the mean.

**Figure 4 ijms-21-09553-f004:**
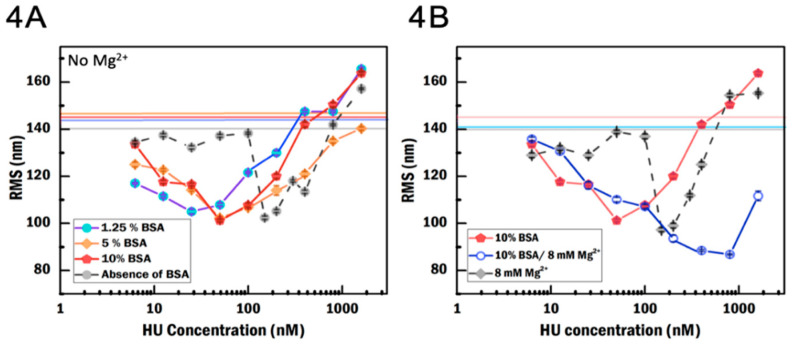
TPM measurements of the effect of bovine serum albumin (BSA) and MgCl_2_ on the binding of HU to DNA. (**A**) RMS as a function of HU concentration in different crowding conditions. Horizontal lines represent the RMS of bare DNA in the absence of BSA (gray), 1.25% BSA (blue), 5% BSA (orange), and 10% BSA (red) in the absence of MgCl_2_. (**B**) RMS as a function of HU concentration in different crowding conditions and MgCl_2_ concentrations. Horizontal lines represent the RMS of bare DNA in the absence of BSA and in the presence of MgCl_2_ (gray), with 10% BSA in the absence of MgCl_2_ (pink) and with 10% BSA in the absence of MgCl_2_ (light blue). The second population in RMS both in the absence and presence of MgCl_2_ is removed in (**A**,**B**). Error is the standard error of the mean.

**Figure 5 ijms-21-09553-f005:**
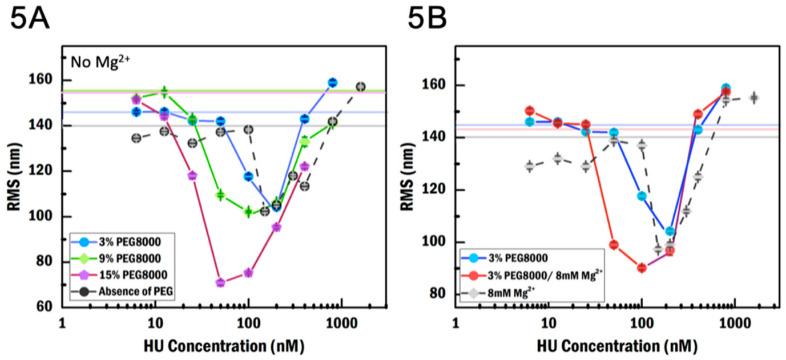
TPM measurements of the effect of PEG8000 and MgCl_2_ on binding of HU to DNA. (**A**) RMS as a function of HU concentration in different crowding conditions. Horizontal lines represent the RMS of bare DNA in the absence of PEG8000 (gray), 3% PEG8000 (blue), 9% PEG8000 (green), and 15% PEG8000 (pink) in the absence of MgCl_2_. (**B**) RMS as a function of HU concentration in different crowding conditions and MgCl_2_ concentrations. Horizontal lines represent the RMS of bare DNA in the absence of PEG8000 and in the presence of MgCl_2_ (gray), with 3% PEG8000 in the absence of MgCl_2_ (pink) and with 10% BSA in the absence of MgCl_2_ (light blue). The second population in RMS both in the absence and presence of MgCl_2_ is removed in (**A**,**B**). Error is the standard error of the mean.

**Figure 6 ijms-21-09553-f006:**
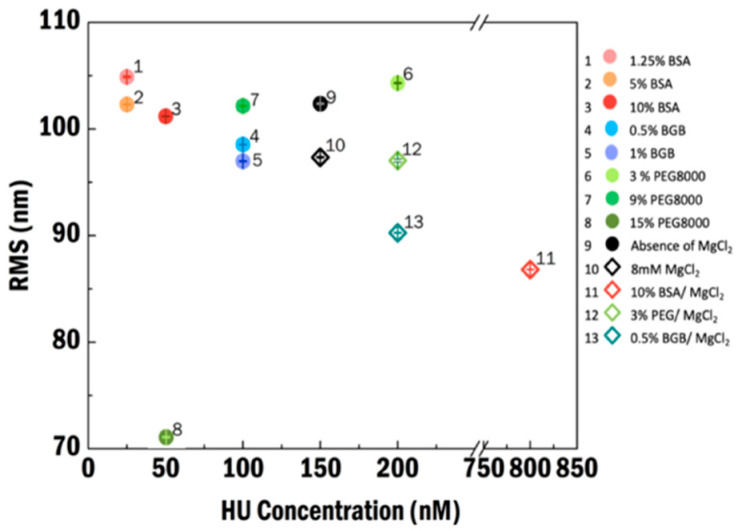
The most compact state in all experimental conditions. Experimental conditions are labeled 1–13. Numbers 1 to 3 are in the presence of BSA in 1.25, 5, and 10% (*v*/*w*) concentration; 4 and 5 are in the presence of 0.5 and 1% (*v*/*w*) BGB; and 6 to 8 are in the presence of PEG8000 in 3, 9, 15% (*v*/*w*), respectively. Both 9 and 10 are crowder-free and in the absence and presence of MgCl_2_, respectively, while 11 to 13 with the empty symbols represent MgCl_2_ conditions with the addition of 0.5% BGB, 10% BSA, and 3% polyethylene glycol 8000 (PEG8000), respectively. Error is the standard error of the mean.

**Figure 7 ijms-21-09553-f007:**
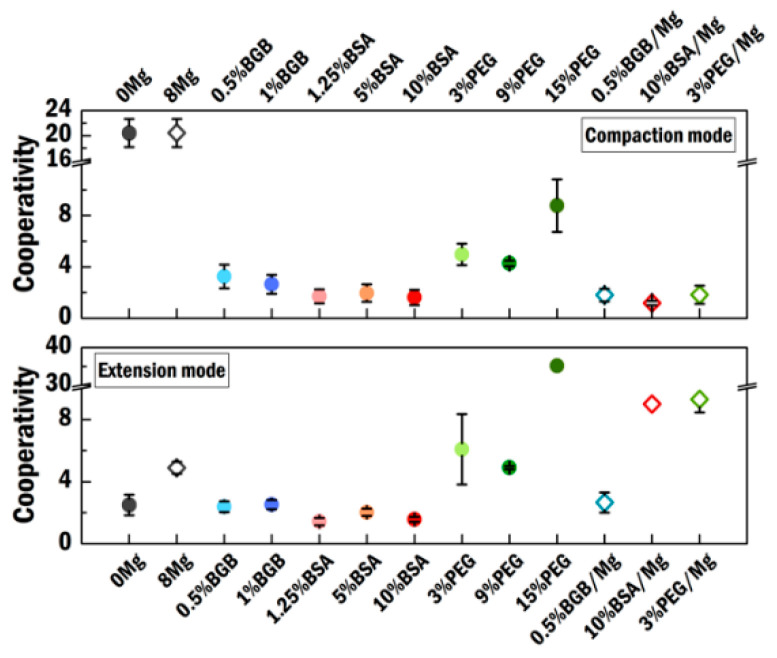
Binding cooperativity in both the compaction and extension regimes. Values are obtained by fitting with the Hill function (Materials and Methods). Values at 0 and 8 Mg^2+^ in compaction mode are shown for reference. Each Mg label represents the experimental condition containing MgCl_2_. Error is the standard error of the mean.

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
