# Peer review of "Effect of Different Crowding Agents on the Architectural Properties of the Bacterial Nucleoid-Associated Protein HU"

_ijms, 2020, doi:10.3390/ijms21249553_

Round 1
Reviewer 1 Report
Lin et al study the effects of crowding agents on the condensation properties of the HU protein on DNA. Many protein-protein or protein-DNA interactions are normally studied in isolated in vitro systems. It is always unclear how those processes differ from their counterparts within the cell. Crowding agents (such as BSA and PEG) are commonly used to generate more ‘cell-like’ conditions that may serve to extrapolate the results of in vitro studies to elucidating in vivo processes. Lin et al use three different crowding agents to study their effect on DNA compaction by HU, a common bacterial protein. The methodology and results are sound and need no further comment. The motivation and interpretation of results is less so. Although the experiments and findings are novel and the interest they will raise is modest as best, the work is technically well performed and I only have minor comments.
I have the following comments:
- [Introduction] I am aware of the use of BSA and PEG is well documented in literature. The authors do not provide any reference on BGB as a crowding agent nor explain what differences they would expect with this particular agent.
- [Materials and Methods] I guess the TPM experiments require a microscope and image analysis, none of which are described.
- [Results] L175 Authors report a 5% lower RMS between conditions. Given the SD of the results, is this difference significant? A similar change of ~4% is described in p6 L206.
- P5 L179-185 Authors describe the presence of two populations. This is potentially an interesting and unexpected result and it raises more questions such as does these two populations exist on the same DNA molecule? Or do some DNA molecules get a HU filament while others don’t? Can these populations interconvert? Why are they observed at only those HU concentrations? However, the topic it is not explored further. In any case, the authors speculate that it is due to HU-DNA fragments that do not cover the entire length of the DNA and may lead to a coexistence of structurally different states. Since it is only speculation it would be more appropriate to move this part to discussion.
- P6 L225 Authors mention that Mg and BGB act synergistically, whereas Figure 3B shows that at low HU concentration compaction is lower in the presence of Mg than in its absence. The authors have the same claim later for Figure 4. Both points should be addressed.
- P7 L245 The conclusion “BSA increases the local concentration of HU” needs to be reconsidered: given the numbers provided in L243-244 one needs 2 times more HU at higher % BSA than at lower % BSA.
- P7 L257 Authors mention that “BSA can potentially interact with DNA”, a reference should be provided to support that statement.
- [Discussion] It seems to me the main finding is that crowding prevents filamentation. This would serve to understand that inside the cell, where large concentrations of HU are present the effect of HU is to compact DNA. If HU was forming filaments inside the cell then no compaction would be observed. In general, neither the introduction nor the discussion put the results of HU and crowding back into a physiological setting. The authors should address this.
- [Discussion] Authors could acknowledge cell extracts as another generic crowding agent for in vitro experiments commonly used to mimic ‘cell-like’ conditions.
Reviewer 2 Report
The manuscript "Effect of different crowding agents on the architectural properties of the bacterial nucleoid-associated protein HU" by Lin SN et al. investigates the HU DNA binding properties in presence of crowding agents and in different salt conditions mimicking cellular environment.
The manuscript is good for novelty and is very interesting because it is a starting point for our understanding about the effect of different cellular conditions on the binding to DNA of HU protein and the DNA structuring state.
The study is well designed and carried out.
As it follows, minor revisions:
In matherials and methods:
- paragraph 2.1 (line 99-100), can the authors detail the conditions of HU overexpression and purification ?
- paragraph 2.2 (line 103), can the authors provide a map of pRD118 in the text or also in Supplementary file ?
This reviewer would also suggest some minor editing of the text hoping to be a help for authors:
-Abstract, line 17-18: the authors write: [...] DNA binding properties of HU under conditions approximating physiological conditions"
The last word "conditions" could be replaced by "ones".
-Introduction, line 47-48: the authors write: Macromolecular crowding effects arise due to macromolecules occupying a significant fraction in the confined volume[...]. Could it be better to write "Macromolecular crowding effects arise because of macromolecules occupancy of a significant fraction..." ?
-Introduction, line 55: the authors write: " That study reveal that that the binding cooperativity of H-NS proteins was increased [17]." The sentence needs to be revised in "The study reveals that..."
-Introduction, line 64-65: the authors write: "Inspired by this earlier study as well as by indications from others studies that different crowding agents have different effects on protein binding [22]..." The sentence needs to be revised in "Inspired by this earlier study as well as by indications from others studies showing that different crowding agents have diverse effects on protein binding [22]...
-Results: at line 167, it could be better to eliminate "(Materials and methods)"
-Results, line 180: it might be better to avoid repeating "HU" after reporting the concentration range.
- Results, line 183, add a coma after "cooperatively"
-Figure legend 6, line 372: eliminate the full stop after " Erros is"
-Discussion and conclusions, line 403: change "That Crowing agents..." with "the crowing agents..."
-Discussion and conclusions, line 410: change "the crowded interior of the cell..." with "the crowded cellular environment..."
-Discussion and conclusions, line 419: change "...but care needs to be taken that..." into " ...but particular care is needed about the effect on the biological functionality of the molecules studied."
